# A Novel HGF/SF Receptor (MET) Agonist Transiently Delays the Disease Progression in an Amyotrophic Lateral Sclerosis Mouse Model by Promoting Neuronal Survival and Dampening the Immune Dysregulation

**DOI:** 10.3390/ijms21228542

**Published:** 2020-11-12

**Authors:** Antonio Vallarola, Massimo Tortarolo, Roberta De Gioia, Luisa Iamele, Hugo de Jonge, Giovanni de Nola, Enrica Bovio, Laura Pasetto, Valentina Bonetto, Mattia Freschi, Caterina Bendotti, Ermanno Gherardi

**Affiliations:** 1Laboratory of Molecular Neurobiology, Department of Neuroscience, Istituto di Ricerche Farmacologiche Mario Negri IRCCS, 20156 Milan, Italy; antonio.vallarola@unimore.it (A.V.); massimo.tortarolo@marionegri.it (M.T.); robertadegioia1989@gmail.com (R.D.G.); mattia.freschi14@gmail.com (M.F.); 2Immunology and General Pathology Unit, Department of Molecular Medicine, Università di Pavia, 27100 Pavia, Italy; luisa.iamele@unipv.it (L.I.); hugo.dejonge@unipv.it (H.d.J.); enrica.bovio@unipv.it (E.B.); egherard@unipv.it (E.G.); 3Department of Neurobiology, Harvard Medical School, Boston, MA 02115, USA; gdenola@crystal.harvard.edu; 4Laboratory of Translational Biomarkers, Department of Biochemistry and Molecular Pharmacology, Istituto di Ricerche Farmacologiche Mario Negri IRCCS, 20156 Milan, Italy; laura.pasetto@marionegri.it (L.P.); valentina.bonetto@marionegri.it (V.B.)

**Keywords:** amyotrophic lateral sclerosis, SOD1G93A mice, motor neurons, HGF/SF, MET, pERK, T cells

## Abstract

Amyotrophic Lateral Sclerosis (ALS) is a fatal neurodegenerative disease with no effective treatment. The Hepatocyte Growth Factor/Scatter Factor (HGF/SF), through its receptor MET, is one of the most potent survival-promoting factors for motor neurons (MN) and is known as a modulator of immune cell function. We recently developed a novel recombinant MET agonist optimized for therapy, designated K1K1. K1K1 was ten times more potent than HGF/SF in preventing MN loss in an in vitro model of ALS. Treatments with K1K1 delayed the onset of muscular impairment and reduced MN loss and skeletal muscle denervation of superoxide dismutase 1 G93A (SOD1G93A) mice. This effect was associated with increased levels of phospho-extracellular signal-related kinase (pERK) in the spinal cord and sciatic nerves and the activation of non-myelinating Schwann cells. Moreover, reduced activated microglia and astroglia, lower T cells infiltration and increased interleukin 4 (IL4) levels were found in the lumbar spinal cord of K1K1 treated mice. K1K1 treatment also prevented the infiltration of T cells in skeletal muscle of SOD1G93A mice. All these protective effects were lost on long-term treatment suggesting a mechanism of drug tolerance. These data provide a rational justification for further exploring the long-term loss of K1K1 efficacy in the perspective of providing a potential treatment for ALS.

## 1. Introduction

The last decade has witnessed unprecedented progress in understanding the genetic complexity of amyotrophic lateral sclerosis (ALS), the most common motor neuron disease, and it is now clear that regardless of the gene mutation, ALS results from both cell autonomous degeneration of cortical and/or spinal motor neurons as well as non-cell autonomous cell damage [1].

Evidence from ALS mouse models, in particular the superoxide dismutase 1 (SOD1) mutant transgenic mice, together with findings from ALS patients, indicate that multiple mechanisms are involved in the death of the motor neurons (MNs) and the loss of muscular innervation. They include a mix of molecular processes (e.g., protein aggregation, impaired RNA processing and metabolism, mitochondrial dysfunction, oxidative stress, cytoskeletal alterations and impairment of axonal transport) that cause cell-autonomous motor neuron dysfunction [2]. Other mechanisms involve the participation of different cell types of the central nervous system (CNS) such as reactive astroglia and microglia [3], Schwann cells of the peripheral nervous system (PNS) [4] and macrophages and T-cells of the peripheral immune system [5,6] that may also elicit neuroinflammatory processes contributing actively to motor neuron degeneration. However, motor neurons become vulnerable to these mechanisms in adult life through activation of signaling pathways promoting cell death [7] or loss of compensatory pro-survival mechanisms such as Protein kinase B (AKT) or the extracellular signal-related kinase (ERK) signaling pathway [8,9] and/or the loss of an anti-inflammatory protective environment around damaged motor neurons [10].

The hepatocyte growth factor/scatter factor (HGF/SF), through the activation of its receptor MET, is one of the most potent survival-promoting factors for sensory and motor neurons during the development and in adulthood in the event of tissue damage [11,12]. Interestingly, certain residual anterior horn cells from post-mortem ALS patients overexpressed both HGF/SF and MET in comparison with those of normal subjects [13]. Most importantly, transgenic overexpression or intrathecal delivery of the growth factor markedly delayed the progression of the disease in SOD1G93A transgenic mice and rats [14,15]. Hence, early clinical trials involving intramuscular administration of plasmid HGF/SF DNA [16] or intrathecal delivery of HGF/SF protein [17] in ALS patients have displayed adequate safety profiles and are now progressing to efficacy studies. However, HGF/SF has a complex multidomain structure consisting of a 69 kDa alpha-chain comprising an N terminal domain (N) and four kringle domains (K) linked via a disulfide bond to a 34 kDa beta-chain consisting of an inactive serine protease homology domain (SPH) (Appendix A) and may be rapidly degraded in the proteinase-rich microenvironment of degenerating motor neurons and activated glial cells typical of ALS, limiting its efficacy. Therefore, fragments of HGF/SF engineered for superior stability, potency and ability to cross the blood–brain barrier may be more suitable for therapy than the natural molecule.

We recently developed and tested in vitro stable and potent recombinant MET agonists based on HGF/SF. They include NK1, a small natural fragment of the growth factor consisting of the N and first kringle domain (K1) of HGF/SF and encompassing the high-affinity MET binding site [18] (Appendix A), and a novel dimeric form of the K1 domain, designated K1K1 [19]. After preliminary studies in vitro, we selected the most potent protein, K1K1, as the molecule of choice to assess the neuroprotective effect in in vitro and in vivo models of ALS. To overcome potential immunogenicity of the human proteins, a mouse version of K1K1 was generated and produced for the in vivo experiments in SOD1G93A transgenic mice. We observed a delay in the disease course of mice treated intraperitoneally with K1K1 and this effect was accompanied by a partial protection of motor neurons and neuromuscular junctions as well as alterations in inflammatory and immune response in the spinal cord and skeletal muscle. The molecular mechanisms involved in these effects of K1K1 were examined.

## 2. Results

### 2.1. Production of Recombinant Proteins with Increased Stability

Native HGF/SF, the MET-agonists NK1 and K1K1 were produced in highly optimized expression systems. NK1 is a natural splice-variant with reduced agonistic activity, while the design of recombinant K1K1 is based on extensive functional and structural studies performed in our laboratories in collaboration with colleagues at the Institute Pasteur in Lille, France [19] and is comprised of two linked HGF/SF kringle 1 domains providing two MET receptor binding sites [19,20] (Appendix A). For therapeutic purposes, K1K1 has several advantages over HGF/SF, one of which is superior stability in buffered solutions such as phosphate buffer saline (PBS) as well as in mammalian cell culture medium. To demonstrate this, purified HGF/SF and K1K1 were diluted to a final concentration of 10 μM in PBS and in RPMI medium containing 10% fetal calf serum (FCS) and incubated at 37 °C. The coomassie analysis of samples taken every week over four weeks shows the gradual degradation of HGF/SF in both conditions after two to three weeks (Appendix A). In contrast, K1K1 does not show degradation under the same conditions (Appendix A). Moreover, measurements on a Tycho NT.6 (Nanotemper) confirmed a steady decrease in fluorescence signal (“brightness” at 330 and 350 nm) for HGF/SF while K1K1 remained stable over a three-week period (Appendix A).

Overall, due to its size and superior stability, K1K1 is clearly optimal for application in vivo. Furthermore, to minimize the chance of K1K1 invoking an immune response during the long-term treatment in mouse, a recombinant mouse variant of K1K1 was generated and used for the in vivo studies. Mouse kringle 1 domain differs from the human kringle 1 in only two amino acid positions: G134 (R134 in human) and N152 (S152 in human). The mouse protein was produced as described for human K1K1 in material and methods [19]. The mouse protein was tested in an MDCK assay and its biological activity was confirmed to be comparable to that of human K1K1 and mouse and human HGF/SF showing observable colony scattering down to the picomolar concentration and differing only by a half-log (3.16×) dilution (Appendix A).

### 2.2. MET Agonists Protects Motor Neurons in Astrocyte-Spinal Neuron Co-Cultures from Transgenic SOD1G93A Mice

In our cellular model of ALS, a spontaneous and selective loss of large mutant motor neurons expressing SOD1G93A transgene occurs after 6 days in vitro (DIV) when compared with non-transgenic cultures under the same conditions [21,22]. To compare the activity of HGF/SF, NK1 and K1K1 in this model, SOD1G93A and non-transgenic co-cultures were treated with vehicle (PBS), HGF/SF or MET agonists, administered at different doses at 0, 2 and 4 DIV. At 6 DIV, cultures were fixed and SMI32-positive MNs counted. Treatment with HGF/SF confirmed its neuro-protective properties, preventing MNs loss in co-cultures expressing SOD1G93A (viable MNs 84.1% ± 12.5, mean ± SEM) only at the highest dose of 10^−9^ M, compared to the untreated transgenic co-culture (viable MNs 43.8% ± 2.0, mean ± SEM) (Figure 1A). K1K1 showed a powerful neuro-protective effect with a total recovery of the MNs viability at a ten-times lower concentration (10^−10^ M, viable MNs 113.4% ± 17.5, mean ± SEM) (Figure 1B), while NK1 reaches similar maximum protective action (as K1K1 at 10^−9^ M) at a ten times higher concentration (10^−8^ M, viable MNs 100.9% ± 11.3, mean ± SEM) (Figure 1C). K1K1 was then used for subsequent in vivo studies.

### 2.3. K1K1 Delays the Onset of Neuromuscular Impairment of SOD1G93A Mice

For the in vivo study, we initially examined the effect of a dose response treatment with K1K1 on the disease progression and the neuropathological alterations in mSOD1 mice. Mice were treated for 6 weeks from the onset of the symptoms (98 days of age) until the symptomatic stage (140 days of age). Female C57BL/6 SOD1G93A mice were randomly divided into four groups and treated intraperitoneally (IP) with PBS as vehicle or mouse K1K1 (0.25, 2.5, and 25 μg/injection) every day, five days a week. The mice were examined for the assessment of neuromuscular deficit twice a week during the entire period of treatment. The mice treated with the higher dose of K1K1 (25 μg) performed the grip strength test significantly better than the vehicle group (Figure 2A), showing a modest but significant delay of the onset of muscle force impairment, evaluated as the age (days) at which the mouse displays the first failure in the grip strength test (K1K1 25 μg: 127.1 ± 1.5 days vs. vehicle 121.6 ± 1.6 days, mean ± SEM) (Figure 2B). No significant changes were observed at lower concentrations of K1K1 (K1K1 0.25 μg: 119.5 ± 1.4 days; K1K1 2.5 μg: 121.9 ± 1.6 days, mean ± SEM).

### 2.4. K1K1 Protects Motor Neurons and the NMJs of SOD1G93A Mice

To evaluate the potential neuroprotective effect of treatments, the number of Nissl stained motor neurons (with an area larger than 400 μm^2^) in the lumbar spinal cord (LSC) was considered for the analysis (Figure 2C). A marked motor neuron loss in SOD1G93A mice treated with vehicle was observed (MNs per hemisection: 5.1 ± 0.7, mean ± SEM) with respect to non-transgenic mice (14.5 ± 1.7, mean ± SEM). Only K1K1 25 μg displayed a partial but significant reduction in motor neuron loss (9.8 ± 0.7, mean ± SEM) (Figure 2D) while no effect was found at the lower doses (Appendix A).

We also evaluated the neuromuscular junction (NMJ) innervation in tibialis anterior muscle (TAM) identified by the co-localization of pre-synaptic (anti-SV2 antibody) and post-synaptic (Bungarotoxin, BTX) terminal staining (Figure 2E). The number of denervated plaques (lack of SV2-BTX co-localization) were calculated as a percentage of the total plaques. SOD1G93A mice treated with vehicle showed a higher percentage (60.1% ± 10.8, mean ± SEM) of denervated plaques than NTG mice (14.1% ± 2.8, mean ± SEM). Treatment with K1K1 25 μg partially but significantly reduced the plaque denervation (32.7% ± 4.8, mean ± SEM) (Figure 2F). The TAM weight was reduced by 51% in SOD1G93A mice treated with vehicle (25.1 ± 5.2 mg; mean ± SEM) compared to NTG (51.6 ± 3.3 mg) but was not significantly modified by the treatment with K1K1 (31.8 ± 2.5 mg; a reduction of 38%). No effect on denervated plaques was found at the lower doses of K1K1 (Appendix A).

### 2.5. K1K1 Activates p-ERK/ERK but not p-AKT/AKT in the Spinal Cord of SOD1G93A Mice

The two most important signaling cascades that the HGF/SF–MET axis is able to trigger are the mitogen-activated protein kinase (MAPK)/extracellular signal-regulated kinase (ERK) and PI3K–AKT pro survival pathways [23]. The activation of both these pathways have been shown to counteract the motor neuron demise in SOD1G93A mice [8,9]. To investigate whether the recombinant fragment K1K1 is capable of stimulating the same mechanisms, the activation of these pathways in the ventral horn of LSC of SOD1G93A mice was analyzed. As shown in Figure 3A, the chronic treatment with K1K1 was unable to increase the ratio between pAKT and AKT, when compared to vehicle treated C57BL/6-SOD1G93A mice. On the contrary, the ratio between phosphorylated ERK (p-ERK) and total ERK was found significantly increased in the spinal cord of SOD1G93A mice treated with K1K1 compared with vehicle treated mice (Figure 3B). Levels of mutant SOD1 were unchanged by the K1K1 treatment (Appendix A).

### 2.6. Effect of K1K1 on the Sciatic Nerve Dysfunction in SOD1G93A Mice

Alterations of Schwann cells (SCs) and consequent demyelination of peripheral nerves have been implicated in the disease progression in SOD1G93A mice [22,24]. Since HGF/SF and MET have been reported to play important roles in SCs-mediated nerve repair by activating the ERK pathway [23], we examined whether K1K1 may alter the activation of ERK and the expression level of Glia Acid Fibrillary Protein (GFAP) in the sciatic nerves of SOD1G93A mice, involved in the de-differentiation and proliferation of SCs, respectively [23,25]. We also examined the levels of acetyl-tubulin as index of axonal transport which was reported to be reduced in the sciatic nerve of SOD1G93A mice [24].

Figure 4 shows that the levels of p-ERK (A) and GFAP (B) were increased in the sciatic nerve of SOD1G93A mice treated with K1K1 with respect to the vehicle treated mice. On the contrary, the levels of acetyl-tubulin were markedly reduced in SOD1G93A mice compared to non-transgenic littermates and were not affected by K1K1 treatment (C).

### 2.7. K1K1 Dampens the Immune Dysfunction in the Lumbar Spinal Cord and in Tibialis Anterior Muscles of SOD1G93A Mice

To evaluate the effect of the K1K1 treatment on the neuroinflammatory response in the spinal cord, we measured the astrocyte and macrophagic microglia activation by immunohistochemical analysis of the Glia Acid Fibrillary Protein (GFAP) and Cluster of Differentiation 68 (CD68) expression, respectively. As seen in Figure 5, the increase in GFAP (A) and CD68 (B) immunostaining observed in SOD1G93A mice treated with the vehicle was significantly reduced by the treatment with K1K1. However, the transcript levels of pro-inflammatory cytokines Tumor Necrosis Factor (TNF)α and (C-C motif) ligand 2 (CCL2) that were significantly increased in SOD1G93A mice, were not modified by treatment with K1K1 (Figure 5C,D). On the contrary, the mRNA expression of anti-inflammatory interleukin 4 (IL-4) was significantly increased in K1K1 treated mice (Figure 5E).

As HGF/SF has been reported to exert immunosuppressive properties on both CD4^+^ and CD8^+^ T cells in a mouse model of multiple sclerosis [26,27] and growing evidence suggest a role of the adaptive immune system in the pathogenesis and progression of ALS [10,24], we investigated the effect of K1K1 treatment on this mechanism in our ALS model. We measured the transcript levels of markers for CD8^+^ and CD4^+^ T cells as markers of cytotoxic and helper T lymphocytes, respectively. An increase in both transcript levels for CD8 and CD4 was observed in the LSC of SOD1G93A mice at the symptomatic stage compared to non-transgenic littermates, and this effect was significantly counteracted by the treatment with K1K1 (Figure 5F,G). On the contrary, the level of forkhead box P3 (FOXP3) mRNA, a key element for T regulatory (Treg) CD4 cell function, was unchanged between SOD1G93A treated with vehicle or K1K1 (Figure 5H). As result the ratio between the transcript levels of FoxP3 (Treg) and those of total CD4^+^ T was higher in SOD1G93A treated with K1K1 (1.04 ± 0.25, mean ± SEM) than in vehicle treated mice (0.39 ± 0.14, mean ± SEM).

Since the immune–inflammatory environment plays a key role in the skeletal muscle regeneration [28], we examined whether K1K1 treatment modified the expression levels of T lymphocytes and macrophages in the TAM of SOD1G93A mice. As shown in Figure 6, the levels of CD8 (A) and CD4 (B) that significantly increased in the TAM of SOD1G93A mice compared to NTG mice, were reduced by the treatment with K1K1 although only for the CD8 the effect reached the significance. Furthermore, CD68 and TNFalpha mRNA levels showed a similar trend (Figure 6C,D), even if the effect was not statistically significant. Overall, these data suggest a prominent inhibitory effect of K1K1 on the immune activation induced by mutant SOD1.

### 2.8. K1K1 Does Not Affect the Onset of Paralysis and Survival of C57BL/6 SOD1G93A Mice

Based on the promising results described above the long-term effect of K1K1 was examined in a new, larger group of SOD1G93A mice treated with the same protocol until the end stage. In this new group of mice, we confirmed that the onset of grip strength impairment was significantly delayed by the treatment with 25 μg K1K1 (125.9 ± 2.1 days vs. vehicle: 119.7 ± 1.5 days, mean ± SEM) (*p* < 0.05) (Figure 7A). However, there was no effect on the age of paralysis (*p* = 0.2956) (Figure 7B), and on the survival (*p* = 0.1048) of SOD1G93A mice (Figure 7C). The disease progression appeared accelerated after 20 weeks of age resulting in a trend of reduced survival of the mice treated with K1K1 with respect to vehicle treated mice (25 μg K1K1: 167.4 ± 2.3 days vs vehicle: 172.9 ± 3.2 days, mean ± SEM). We therefore examined whether the prolonged treatment with K1K1 was still able to activate the ERK pathway in the spinal cord and the sciatic nerve and was able to counteract the immune cell infiltration in the spinal cord and TAM at the end stage of the disease. We found that the increase in *p*-ERK levels in both spinal cord and sciatic nerve observed in SOD1G93A mice treated for 6 weeks with K1K1 was abolished after prolonging the treatment four more weeks up to the end stage of the disease (Appendix A). Similarly, the increased levels of mRNA for CD8^+^ and CD4^+^ T cells in the spinal cord and TAM of SOD1G93A mice, compared to non-transgenic littermates, were not reduced during long-term treatment with K1K1 (Appendix A).

## 3. Discussion

The present study strengthens the evidence that activation of the MET receptor by engineered derivatives of HGF/SF results in a strong neuroprotective activity on the motor neurons of SOD1G93A transgenic mice and demonstrates for the first time—to the best of our knowledge—that this result can be accomplished through the systemic administration of a potent MET ligand. Treatment with K1K1 resulted in the rescue of motor neurons from death in spinal-neuron astrocyte co-cultures as well as in the lumbar spinal cord of SOD1G93A transgenic mice accompanied by a transient amelioration of the muscle force impairment, although it did not prolong the mouse survival. This improved outcome of K1K1 treated ALS mice was characterized by the protection of motor neurons both at the level of their soma in the spinal cord and of the motor axons terminals as demonstrated by the reduced neuromuscular junction denervation. In the spinal cord, we found that the neuroprotection was accompanied by the activation (phosphorylation) of ERK but not AKT. This result complies with a recent study showing that intrathecal delivery of recombinant AAV1 encoding HGF/SF protected spinal motor neurons in SOD1G93A mice through an increase in phosphorylated ERK but no other signaling molecules of the HGF/SF-MET pathway such as STAT3, cJUN and GSK3b [15]. This highlights the importance of ERK activation in the rescue of motor neurons induced by HGF/MET signaling. Interestingly, ERK was remarkably activated also in the sciatic nerves of SOD1G93A mice treated with K1K1 together with the increased expression of GFAP. Activation of ERK and GFAP in the sciatic nerve is essential to stimulate the de-differentiation and promote the proliferation of Schwann cells, respectively, during re-innervation [25]. This is consistent with the recent evidence demonstrating that HGF/SF increased the migration and proliferation of cultured Schwann cells by activating the ERK pathway and accelerated the nerve regeneration process after nerve crush [29]. Interestingly, we previously demonstrated that the expression of GFAP and p-ERK were higher in the sciatic nerves of SOD1G93A mice showing a delayed disease onset and progression compared to those with fast disease progression and this was associated with higher levels of acetyl-tubulin, a marker of axonal function [24]. With K1K1 treatment, we did not find a rescue of acetyl-tubulin reduction in symptomatic SOD1G93A mice, however it should be noted that while in the previous study the analysis had been performed at the disease onset, here the mice were examined at the symptomatic stage. We cannot rule out that a rescue of acetylated tubulin occurred at an earlier time point after K1K1 treatment. Noteworthily, the prolonged treatment with K1K1 until the end stage of the disease lost its efficacy in activating p-ERK/ERK in LSC and sciatic nerve of SOD1G93A mice (Appendix A) suggesting a mechanism of “drug tolerance” which could explain the transiency of the amelioration of the neuromuscular deficit of SOD1G93A mice.

K1K1 treatment also produced a significant reduction in the reactive astrogliosis and microgliosis in the LSC of SOD1G93A mice. We do not know whether this is a response to the partial rescue of MNs or is causative of the neuroprotection. Unexpectedly, the reduction in reactive gliosis did not match with a reduction in proinflammatory cytokines TNFalpha and CCL2 that remained higher in SOD1G93A mice after treatment with K1K1. However, we detected a concomitant increase in anti-inflammatory IL-4 in the LSC of K1K1 treated mice, suggesting that reactive microglia, besides being reduced in number, might have adopted a partial anti-inflammatory phenotype in response to K1K1 treatment [30]. An association between motor neuron protection, reduction in microgliosis and increased IL-4 was previously observed with other treatments in SOD1G93A mice sustaining the protective role of this cytokine [22,31]. Interestingly, the intracerebroventricular delivery of IL-4 in SOD1G93A mice, via a lentiviral-mediated gene therapy strategy, was able to modulate microglia and to delay the disease onset in these mice but did not prolong their survival [32].

Another potential mechanism through which K1K1 treatment may exert it protective effect on neuromuscular system relates to the modulation of the immune response in both the spinal cord and the skeletal muscle. The MET signaling pathways have been implicated in the modulation of different immune–inflammatory responses [33]. For example, in a multiple sclerosis mouse model HGF/SF has been shown to modulate both CD4^+^ and CD8^+^ T cell effector responses [26,27]. Here, we show that the increased recruitment of CD8^+^ and CD4^+^ T cells in the spinal cord of symptomatic SOD1G93A mice was significantly counteracted by treatment with K1K1 during the early progressing phase of the disease, although this effect disappeared on the long-term treatment. Whether this effect may depend on a direct action of K1K1 on the migratory activity of T cells or being a consequence of the protective effect on the MNs needs to be investigated. We and other groups have recently demonstrated that the ablation of CD8^+^ T cells in SOD1G93A mice protected spinal motor neurons from death at the early symptomatic stage [24,34] and this effect was associated with the reduction in macrophagic microglia hyperactivation [24]. On the contrary, the lack of CD4^+^ T cells in SOD1G93A mice showed a detrimental effect on disease progression, in the presence of attenuated microglia and astrocyte reactivity in the spinal cord [35]. The CD4^+^ T lymphocytes in the spinal cord of SOD1G93A mice are probably a mixture of Th1 and Th2 effector T cells (Teffs) and Th2 lymphocytes possibly provide the increased, although not significant, levels of IL-4 mRNA found in vehicle treated SOD1G93A mice, compared to NTG. However, IL-4 mRNA levels were further increased by the treatment with K1K1 despite the reduction in CD4^+^ T cells mRNA in the spinal cord. IL-4 is also produced by the Treg, a CD4^+^ T lymphocyte subtype with marked neuroprotective effect in SOD1G93A mice [32]. Through the analysis of FoxP3, a transcription factor typical of the Treg, we showed that the proportion of Tregs present in the CD4^+^ T cell population of SOD1G93A mice treated with K1K1 is higher than that present in vehicle treated mice. This is consistent with the protective action of K1K1 in SOD1G93A mice. In fact, growing evidence suggest that the expansion of Tregs plays a significant role in the modulation of disease progression in ALS patients and mouse models [6,10]. Passive transfer of T-cell populations enriched in Tregs were shown to sustain IL-4 levels and M2 microglia, lengthen disease duration, and prolong survival of SOD1G93A mice [35]. Moreover, treatment of transgenic mice with interleukin 2 complex (IL-2c) with rapamycin, which is known to induced Treg expansion, significantly increased the levels of FOXP3 mRNA in the spinal cord of SOD1G93A mice and this effect was associated to a rescue of motor neurons and reduced glial activation [36]. It has been reported that dendritic cells exposed to HGF/SF develop tolerogenic properties and facilitate the expansion of Tregs [37]. Dendritic cells (DCs) were detected early in the spinal cord of SOD1G93A mice [38] as well as in post-mortem ALS patient spinal cord [39]. Thus, we can speculate that infiltrated DCs could have been activated by K1K1 in the spinal cord of SOD1G93A mice in an attempt to preserve the Treg population.

The immunomodulatory effect by K1K1 in SOD1G93A mice was detected at the skeletal muscle level as well. In fact, the overexpression of macrophages and T lymphocytes found in the TAM of symptomatic SOD1G93A mice were markedly reduced in K1K1 treated mice with consequent partial reduction in the proinflammatory cytokine, TNFalpha. There is a consolidated hypothesis that, following damage, the skeletal muscle activates a series of immune mediators and proinflammatory cytokines that allow the recruitment of macrophages, neutrophils and adaptive immune T cells important for the improvement and modulation of the muscle growth and regeneration [28]. We recently reported that a depletion of CD8^+^ T cells in SOD1G93A mice lacking beta2-microglobulin, while protecting the soma of MNs in spinal cord, accelerated the NMJs denervation in the TAM and this was accompanied by an anticipated onset of hind limb impairment [24]. However, the effect was transient as at the later symptomatic stage there was no difference in TAM NMJs denervation between SOD1G93A mice with or without CD8^+^ T cells while the onset of disability was delayed [24]. Apparently, this result is at variance with those obtained in the present study showing that the decrease in CD8^+^ T induced by K1K1 was accompanied by reduced NMJs denervation. Nevertheless, unlike the study on counteracting roles of MHCI and CD8^+^ T cells [24], with K1K1 we induced a massive immune suppression in the skeletal muscle that include both the macrophages and the CD4^+^ T cells and reduced the proinflammatory cytokine TNFalpha that may have contributed to the delay of symptoms. Activated macrophages have been reported to infiltrate and accumulate in the skeletal muscles of SOD1G93A mice beginning from disease onset [39] and its reduction by ablation of complement signaling was reported to reduce NMJs denervation and to improve hind limb grip strength in mice [40] similarly to what we observed with K1K1. Noteworthily, although we found that K1K1 treatment partially rescued the denervation of NMJs in TAM, the decrease in muscle mass was not counteracted by the treatment. Since there is evidence that the recruitment of CD4^+^ T cells in damaged skeletal muscles is important for the repair and the regeneration of the muscle [28], we cannot exclude that their reduction by K1K1 may have interfered with this process.

## 4. Materials and Methods

The wild-type human and mouse HGF/SF were produced in the myeloma line NS0 (MRC, UK) and purified as previously described [18]. The natural splice-variant NK1 was produced recombinantly in the yeast expression system *Pichia pastoris* (Invitrogen, Carlsbad, CA, USA) [41]. The engineered recombinant K1K1 was produced as described in Leclercq et al., 2020 [19]. Briefly, K1K1 was produced in bacterial inclusion bodies using the *E. coli* strain BL21 (Invitrogen) giving yields of around 10 mg/litre. The protein was extracted from the inclusion bodies using a Tris buffered 2 M arginine solution and was subsequently diluted in a Tris pH 7.4 buffer and purified by Heparin Sepharose affinity and gel filtration chromatography. Each batch of MET agonist was quantified by absorbance measurement at 280 nm or standard BCA assay (Pierce, Walthman, MA, USA).

### 4.1. Madin-Darby Canine Kidney Cell Colony Scatter Assay

For the testing of biological activity, mouse and human variants of HGF/SF and K1K1 were tested in a sensitive colony dispersal (“scatter”) assay [18]. MDCK cells (ICRF, UK) were maintained in DMEM (Gibco, Life Technologies, Grovemont, MD, USA) supplemented with 10% FCS (Gibco) in a humidified incubator at 37 °C with 5% CO_2_. Small and compact colonies appear overnight after seeding 1000 cells per well in a 96-well plate. The medium was replaced with DMEM + 10% FCS supplemented with the ligands at different concentrations ranging from 10 µM down to 0.1 pM using serial half-log dilutions. The next day, colony dispersal was observed at low magnification under an inverted microscope and the endpoint of observable scattering was determined for each protein. Using low magnification, three photos were taken of individual colonies at each concentration for all ligands.

### 4.2. Quantification of Protein Degradation

Purified HGF/SF and K1K1 was diluted to a final concentration of 10 µM in 200 µL Phosphate Buffer Saline (PBS) at pH 7.4 and in 200 µL RPMI medium (Gibco, Life Technology) supplemented with 10% FCS (Gibco). The proteins were incubated at 37 °C for 4 weeks. Samples of 40 µl were collected each week, flash frozen and stored at −80 °C for further analysis by SDS-PAGE and Thermal shift assay. An amount of 5 µg HGF/SF and 3.5 µg K1K1 sample was loaded on a 12% SDS-PAGE gel and subsequently stained with Coomassie Blue. A thermal shift assay was performed using the Tycho NT.6 (NanoTemper Technologies, München, Germany) measuring changes in total fluorescence intensity (“brightness”) at 330 nm and 350 nm with a 30 °C/min ramp (from 35 to 95 °C). The total sample brightness was used for calculating the percentage reduction in soluble protein in the samples using Excel (Microsoft) and the bar chart was produced using Prism (Graphpad, San Diego, CS, USA).

### 4.3. SOD1G93A Mice

Transgenic mice (C57B6.CgTg SOD1.G93A1Gur/J) were originally obtained from Jackson Laboratories (Bar Harbor, ME, USA) and then maintained on a C57BL6/J background at the Istituto di Ricerche Farmacologiche Mario Negri IRCCS, Milan, Italy (IRFMN). The animals were housed under specific pathogen free (SPF) standard conditions (22 °C ± 1, 55% ± 10 relative humidity and 12 h light/dark schedule), 3–4 per cage, with free access to food (standard pellet, Altromin, MT, Rieper) and water. Procedures involving animals and their care were conducted in conformity with the institutional guidelines of the Mario Negri Institute for Pharmacological Research IRCCS, Milan, Italy, which are in compliance with national (D.lgs 26/2014; Authorization n.783/2016-PR issued on 8 August 2016 by Ministry of Health) and Mario Negri Institutional regulations and Policies providing internal authorization for persons conducting animal experiments (Quality Management System certificate—UNI EN ISO 9001:2008–reg. N° 6121), the NIH Guide for the Care and Use of Laboratory Animals (2011 edition) and EU directives and guidelines (EEC Council Directive 2010/63/UE).

### 4.4. Primary Astrocyte-Spinal Neuron Co-Cultures

SOD1G93A and non-transgenic co-cultures were prepared as previously described [21,22]. Briefly, astrocytes were obtained dissecting cortices of E13-E14 embryos from SOD1G93A mice or their non-transgenic littermates and mechanically dissociation in Hanks’ balanced salt solution (HBSS) containing 33 mM glucose. After centrifugation, the pellet was suspended in culture medium (Dulbecco’s modified Eagle’s medium/F12, 2 mM L-glutamine, 33 mM glucose, 5 µg/mL gentamycin, 10% horse serum) and seeded (500,000 cells/mL) onto 48- or 6-well plates coated with 1.5 µg/mL poly-L-ornithine. Repeated washing with HBSS, 12 h of orbital shaking at 200 rpm and treatment with 10 M AraC once they reached confluence, rendered astrocyte cultures free of microglia, oligodendrocytes, and neurons. Spinal cords of E13-E14 embryos were dissected and mechanically dissociated in HBSS, 33 mM glucose. The cells were centrifuged onto a 4% bovine serum albumin cushion at 201 rcf for 10 min and the pellet resuspended in neuron culture medium: Neurobasal (Gibco), 2 mM L-glutamine, 33 mM glucose, 5 µg/mL gentamycin, 1 ng/mL brain-derived neurotrophic factor, 25 µg/mL insulin, 10 µg/mL putrescine, 30 nM sodium selenite, 2 µM progesterone, 100 µg/mL apo-transferrin, 10% heat-inactivated horse serum, 10 µM AraC. Cells were seeded (1,000,000 cells/mL) onto 48- or 6-well plates coated with 15 µg/mL poly-L-ornithine and 2 µg/mL laminin or onto a pre-established astrocyte confluent layer to obtain spinal neuron-cortical astrocyte co-cultures. Non-transgenic and SOD1G93A co-culture were obtained from non-transgenic and SOD1G93A neurons seeded on non-transgenic and SOD1G93A astrocytes, respectively. Motor neurons obtained from E13–14 embryos are fully differentiated and express the specific transcription factors Hb9 and Islet-1/2. A few days after plating, they show adult characteristics such as profuse dendrite and axon outgrowth and synapse formation [21,22]. Co-cultures were treated with HGF/SF, NK1 and K1K1. The molecules were administered at different doses every other day starting from the plating of the neurons, days in vitro (DIV) 0, 2 and 4. Co-cultures were fixed and analyzed at 6 DIV.

### 4.5. Immunocytochemistry and Evaluation of Motor Neuron Survival In Vitro

Immunocytochemistry was performed as previously described [21,22,31]. After an incubation with blocking solution containing 10% normal goat serum (NGS) and 0.1% Triton in phosphate-buffered saline (PBS) 0.01 M, the cells were incubated overnight at 4 °C with the primary antibodies mouse monoclonal anti-SMI32 (1:1000, Covance, Cambridge, MA, USA) and mouse monoclonal anti-NeuN (1:250, Chemicon, Burlington, MA, USA), diluted in a solution containing 1% NGS and 0.1% Triton in PBS 0.01M. Cells were then washed and incubated with the appropriate secondary fluorescent antibody (1:500, Alexa Fluor Dyes, Life Technologies) or secondary biotinylated antibody (1:500, Vector Laboratories, Burlingame, CA, USA) for tyramide signal amplification (TSA, Perkin Elmer, Walthman, MA, USA) following the manufacturer’s instructions. Images were acquired with an Olympus BX41 fluorescence microscope. Motor neuron survival was evaluated as previously described [21,22]. The labelling with anti-SMI32 antibody highlights motor neurons with typical morphology and large cell bodies (diameter ≥ 20 μm) and anti-NeuN antibody was used to identify all neurons in twelve adjacent frames per well at 10X-magnification. Data were calculated as the ratio of the number of motor neurons to the total neurons and expressed as percentage of untreated non-transgenic co-culture to compare different experiments.

### 4.6. MET Agonist K1K1 In Vivo Treatment

For the first in vivo treatment SOD1G93A female mice were treated intraperitoneally (IP) with PBS vehicle or mouse K1K1 (0.25, 2.5, and 25 μg/injection) every day, five days a week, starting from 98 days of age, when the mice show the first signs of symptoms (hind limb tremors and reduced limb abduction), and until the overt symptomatic phase (140 days of age). At 140 days of age, ten mice per group were sacrificed to perform histopathological, biochemical and molecular analyses. For the second in vivo treatment, 15 female SOD1G93A mice per group were treated IP with vehicle or K1K1 (25 μg) every day, five days a week, starting at the onset of symptoms (98 days of age) and until the end stage of the disease to assess the effect on survival.

### 4.7. Behavioural Analysis and Survival

Behavioral analyses were performed in all mice two times a week from the onset of the disease, by the same investigator blinded to the treatment. The grip strength test was used to measure disease progression by evaluation the limb resistance as previously described [21,22,31]. Mice were placed on a horizontal metallic grid which was then gently inverted. The latency to fall of each mouse was recorded. The test ended after 90 s. The measurement was repeated three times in case the mice fell off before 90 s and the best performance holding on the grid was considered for the statistical analysis. The onset of neuromuscular deficit was evaluated at the age when the mouse exhibits the first failure in the paw grip strength test at two consecutive time points. The age at which the mice were no longer able to perform the grip strength test was considered as time of paralysis. The mice were sacrificed when they were unable to right themselves within 10 s after being placed on either side. This time was considered the end stage of the disease and was used to calculate the survival.

### 4.8. Immunohistochemistry

Spinal cords were processed as previously described [22,31]. Briefly, under deep anesthesia, mice were transcardially perfused with PBS followed by 4% paraformaldehyde (PFA) solution. The spinal cords were quickly removed, post fixed for 24 h, and cryopreserved in 30% sucrose solution at 4 °C until they sank, embedded in Tissue-tek OCT (Sakura, AJ Alphen aan den Rijn, The Netherlands), frozen in n-pentane at −45 °C and stored at −80 °C until analysis. Spinal cord immunohistochemistry was performed on free floating sections (30 μm). The number of motor neurons was determined on serial sections (20, one every 10th) from lumbar spinal cord (LSC) segments L2-L5 of 5 mice per group as previously described [22]. The sections were stained with cresyl violet to detect the Nissl substance of neuronal cells. Motor neuron counting was performed at 10× magnification using the free software ImageJ (Windows v8.0, http://imagej.nih.gov/ij/), previously calibrated. Intense Nissl labelled neurons with clear nucleus and nucleolus and an area of the cell body higher than 400 μm^2^ were identified as motor neurons. The number of motor neurons was calculated for each hemisection and the means used for statistical analysis. Immunofluorescence was evaluated on 5–7 coronal spinal cord slices (1 every 10) from LSC per animal. After an incubation with blocking solution containing 3% NGS and 0.1% Triton in 0.01 M PBS, primary antibodies *mouse* anti-GFAP (1:2500, Millipore, Burlington, MA, USA) and rat anti-CD68 (1:200, AbDSerotec, Hercules, CA, USA) and appropriate fluoro-conjugated secondary antibodies (1:500 dilution), Alexa 647 and Alexa 488 (Alexa Fluor^®^ Dyes, Life Technologies) were used. The sections were analyzed under Olympus Fluoview confocal microscope. The quantification of GFAP and CD68 intensity in the ventral horns was carried out using the free software ImageJ by determining the area fraction of fluorescent signal after setting a threshold value within a grey-scale (corresponding to the maximum level of an unstained section background) and considering all the pixels falling over this value as positive. These analyses were executed by the same operator blinded to treatment.

### 4.9. Neuromuscular Junctions (NMJs)

NMJ denervation was detected according to the previously described protocol [22]. Briefly, tibialis anterior muscles (TAM) were dissected from mice transcardially perfused with PBS under deep anesthesia and snap-frozen in isopentane cooled on dry ice. Serial 20 µm cryostat longitudinal muscle sections were collected on poly-Lysine objective slides (VWR International), fixed in chilled acetone for 10 min, incubated in a blocking solution (0.3% Triton, 10% NGS in 0.01 M PBS) for 1 h at 22 °C and left overnight at 4 °C with anti-SV2 primary antibody (1:100, DSHB) in 0.15% Triton, 5% NGS, 0.01 M PBS. Then the sections were incubated with goat anti-rabbit 647 (1:500, Alexa Fluor^®^ Dyes, Life Technologies) secondary antibody and with bungarotoxin (1:500, Invitrogen) conjugated with Alexa Fluor^®^ 488 (Life Technologies). Innervated neuromuscular junctions were identified by the bungarotoxin labelling, totally or partially co-localized with synaptophysin labelling. Endplates marked with bungarotoxin only were considered denervated. Data were expressed as the percentage of the denervated plaques over the total ones counted in 8 adjacent frames per section. Five sections at 20× magnification were analyzed for each mouse. Fluorescence images along the z axis were taken by Olympus confocal microscopy using a 20× objective and z-stacking was performed by using ImageJ software.

### 4.10. Western Blot

Mice were sacrificed according to institutional ethical procedures by decapitation and the spinal cord and sciatic nerve were rapidly dissected. The spinal cord was immediately frozen on dry ice and stored at −80 °C. For each mouse, LSC was longitudinally transected at 50 µm in a cryostat with ventral and dorsal spinal cord sections as separate samples. The resulting cryostat ventral material was homogenized by sonication in ice-cold homogenization buffer (20 mM Tris-HCl pH 7.4, 2% Triton X-100, 150 mM NaCl, 1 mM EDTA, 5 mM MgCl_2_, 10% anhydrous glycerol, protease and phosphates inhibitor cocktail by Roche), centrifuged at 15,700 rcf for 30 min at 4 °C and the supernatants were collected and stored at −80 °C. The sciatic nerves were processed as previously described [24]. Briefly, tissues were powdered in liquid nitrogen, next homogenized by sonication in ice-cold homogenization buffer (20 mM Tris-HCl pH 7.4, 1% Triton X-100, 150 mM NaCl, 1 mM EDTA, 5 mM MgCl_2_, 10% anhydrous glycerol, protease and phosphates from Roche) and centrifuged at 15,700 rcf for 15 min at 4 °C. Equal amounts of total protein homogenates were loaded on polyacrylamide gels and electroblotted onto PVDF membrane (Millipore) as previously described [31]. Membranes were first blocked with 5% BSA (Sigma, St. Luis, MO, USA) in TBS with additional 0.1% Tween (TBS-T) for 1 h at room temperature and then incubated over night at 4 °C with one of the following primary antibodies: mouse monoclonal anti-GFAP (1:1000, Millipore), mouse monoclonal anti β-actin (1:15,000, Chemicon), rabbit monoclonal anti pAKT (1:750, Cell Signaling, Danvers, MA, USA), rabbit anti AKT (1:1000, Cell Signaling), mouse anti pERK (1:1000, Santa Cruz Biotechnology, Dallas, Tx, USA ), rabbit anti ERK (1:1000 Promega), rabbit anti human SOD1 (1:1000, Upstate, Burlington, MA, USA), mouse anti CNPase (1:1000 Chemicon), mouse anti β- III Tubulin (1:1000, Millipore), mouse monoclonal anti-Acetylated Tubulin (1:1000, Sigma Aldrich). Membranes were then washed and incubated with horseradish peroxidase-conjugated anti-rabbit, anti-mouse or anti-rat secondary antibodies (Santa Cruz) and developed by Luminata Forte Western Chemiluminescent horse radish peroxidase (HRP) Substrate (Millipore) on the Chemi-Doc XRS system (Bio-Rad, Hercules, CA, USA). Densitometry analysis was performed with ImageLab (Bio-Rad) software.

### 4.11. Real-Time PCR

Spinal cords and muscles were freshly collected from mice transcardially perfused with PBS under deep anesthesia. All tissues were immediately frozen on dry-ice. Tissue was homogenized and total RNA was extracted using the Trizol method (Invitrogen) and purified with PureLink RNA columns (Life Technologies). RNA samples were treated with DNase I and reverse transcription was performed with High Capacity cDNA Reverse Transcription Kit (Life Technologies). Real-time PCR was performed using the Taq Man Gene expression assay (Applied Biosystems, Foster City, CA, USA) following the manufacturer’s instructions, on cDNA specimens in triplicate, using 1× Universal PCR master mix (Life technologies) and 1× mix containing specific probes for Tumor Necrosis Factor (TNFα, Mm00443258_m1), chemokine (C-C motif) ligand 2 (CCL2, Mm00441242_m1), interleukin 4 (IL4, Mm00445259_m1), interleukin 10 (IL-10, Mm00439614_m1), forkhead box P3 (FoxP3, Mm00475162_m1), cluster of differentiation 4 (CD4, Mm00442754_m1), cluster of differentiation 8 (CD8, Mm01182107_g1), cluster of differentiation 68 (CD68, Mm03047343_m1) and β-actin (Hs01060665_g1) all from Life technologies. Relative quantification was calculated from the ratio between the cycle number (Ct) at which the signal crossed a threshold set within the logarithmic phase of the given gene and that of the reference gene (β actin). Mean values of the triplicate results for each animal were used as individual data for 2-ΔΔCt statistical analysis.

### 4.12. Statistical Analysis

One-way ANOVA was used to compare differences between more than two groups, followed by post hoc Fisher’s least significant difference (LSD) while Student’s “t” test was used for the analysis of two groups. Two-way ANOVA was used for the analysis of dose response effect in NTG and SOD1G93A cell cultures and for the analysis of repeated measures of grip strength test in mice. The Mantel–Cox log rank test was used for comparing motor deficit onset, paralysis and survival between groups.

## 5. Conclusions

In summary, this study reveals a straightforward neuroprotective activity of the K1K1 protein which resulted in a transient reduction in the pathological impact of the SOD1G93A mutation in mice during the early symptomatic stage of ALS. The results suggest that MET activation is able to rescue the damage of the neuromuscular system induced by the SOD1G93A mutation through different central and peripheral mechanisms. They include the modulation of immune cell infiltration in the spinal cord with a reduction in cytotoxic T cells and a prevalent enrichment of the Tregs. This effect which disappears after long term treatment with K1K1 in concomitance with the loss of beneficial effects in the later progressing phase of the disease, further implies that the control of T cells infiltration, particularly the cytotoxic CD8^+^, is crucial to fostering and maintain the MNs and NMJs integrity. Another key process triggered by MET to protect MNs is the activation of the de-differentiation and proliferation of Schwann cells, as demonstrated by the activation of ERK and GFAP, respectively, to promote the regeneration of damaged motor axons and muscle reinnervation. Overall, these data provide a rational justification for further exploring the activity of K1K1 in terms of possible dosage optimization and further understanding of long-term loss of efficacy.

## Figures and Tables

**Figure 1 ijms-21-08542-f001:**
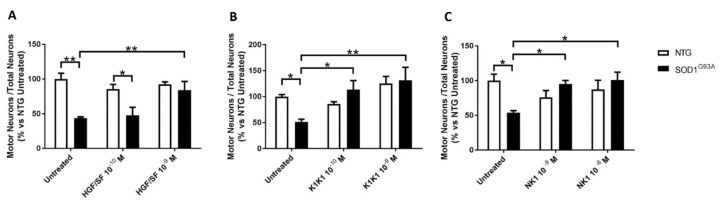
MET agonists protect motor neurons (MNs) from death in a cellular model of Amyotrophic Lateral Sclerosis (ALS). (**A**–**C**) Quantitative assessment of MN survival in non-transgenic (NTG, white) or superoxide dismutase 1 G93A (SOD1G93A) (black) astrocyte-spinal neuron co-cultures after 6 days in vitro (DIV). Cells were treated at DIV 0, 2 and 4 with different doses of hepatocyte growth factor/scatter factor (HGF/SF) or MET agonists or left untreated as controls. The columns show the number of viable MNs calculated as the ratio between the MN number (SMI32-positive, maximum diameter >20 µm) and the number of total NeuN-positive neurons. Data are expressed as a percentage of NTG untreated samples. The analysis of the MNs’ survival revealed that treatment with K1K1 showed the best neuroprotective profile even at a lower dose (10^−10^ M), completely preventing MN loss in transgenic co-cultures. Data are expressed as mean ± SEM (*n* = 5 independent experiments). Data were analyzed with Two-way ANOVA followed by Tukey’s post-hoc Test. * *p* < 0.05, ** *p* < 0.01.

**Figure 2 ijms-21-08542-f002:**
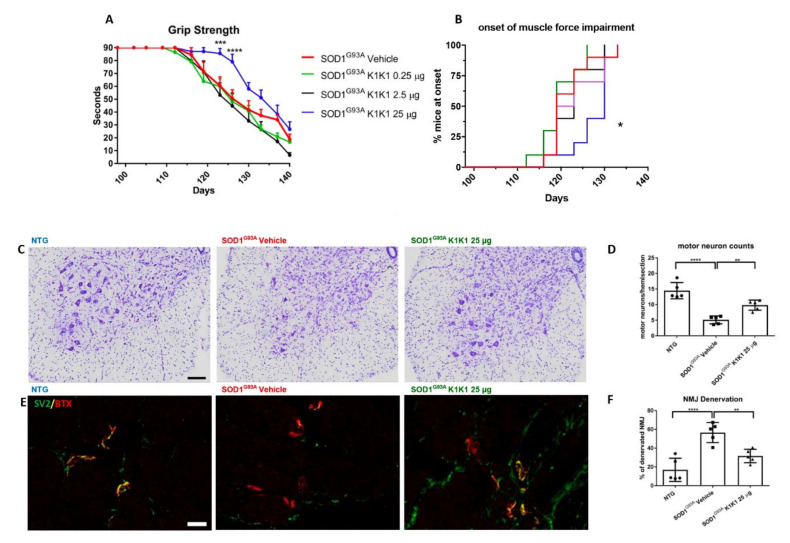
K1K1 25 μg delays neuromuscular impairment and protects the neuromuscular system in SOD1G93A mice. (**A**) A significant improvement in the grip strength test performance is observed in SOD1G93A mice treated intraperitoneally (IP) with K1K1 25 μg but not with lower dosages (0.25 and 2.5 μg) compared to vehicle treated mice. Each point represents the mean ± SEM (*n* = 10). Data were analyzed using Two-way ANOVA for repeated measures (time) and different groups (treatment). (**B**) A Kaplan–Meier curve showed that the onset of motor symptoms was significantly delayed. Data were analyzed using the Log-rank test (*p* < 0.05, *n* = 10 animals for each group). (**C**) Representative images of Nissl staining performed on lumbar spinal cord (LSC) sections from non-transgenic (NTG) and SOD1G93A mice treated with phosphate buffer saline (PBS) (vehicle) or K1K1 25 μg at the symptomatic stage of the disease (140 days). Scale bar: 100 µm. (**D**) Quantification of the motor neuron number (cell body area >400 μm^2^) in lumbar spinal cord. MNs were decreased in transgenic groups compared to NTG and 25 μg K1K1 displayed significant neuroprotection with a partial reduction in motor neuron loss. (**E**) Representative confocal images of synaptic vesicle protein 2 (SV2, green) and bungarotoxin (BTX, red) co-localization in neuromuscular junctions (NMJs) of Tibialis Anterior Muscle (TAM) from NTG and SOD1G93A mice treated with vehicle or K1K1 25 μg (140 days of age). Scale bar: 50 µm. (**F**) Denervated NMJs represented by the lack of co-localization between SV2 and BTX was higher in TAM of transgenic mice treated with vehicle and K1K1 25 μg significantly reduced this effect. (**D**,**F**) Data are expressed as mean ± SEM, *n* = 5 animals per group. Data were analyzed by One-way ANOVA followed by post hoc Fisher’s least significant difference (LSD). * *p* < 0.05, ** *p* < 0.01, *** *p* < 0.005, **** *p* < 0.001.

**Figure 3 ijms-21-08542-f003:**
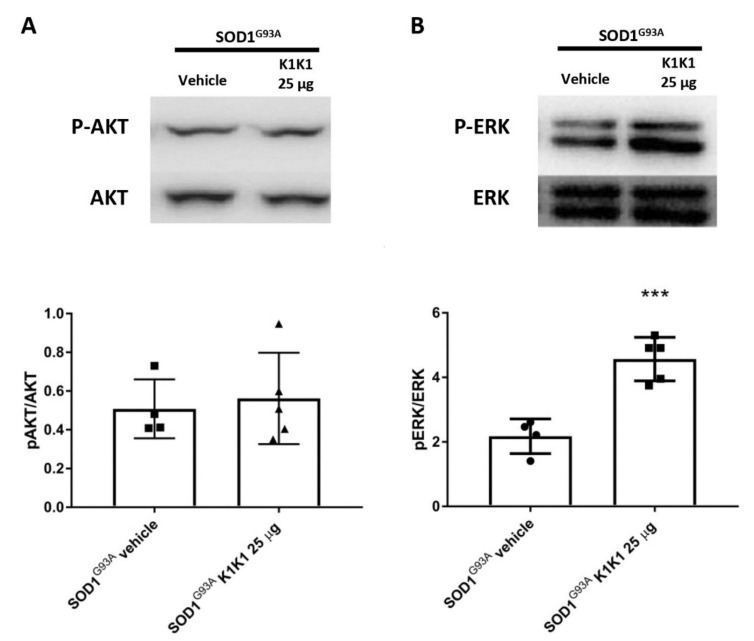
Treatment with 25 μg K1K1 increased only extracellular signal-related kinase (ERK) phosphorylation in lumbar spinal cord of SOD1G93A mice. (**A**,**B**) Representative immunoblots of p- Protein kinase B (AKT)/ AKT and phospho-extracellular signal-related kinase (p-ERK)/ERK and relative quantification performed on the ventral portion of lumbar spinal cord of transgenic SOD1G93A mice treated with vehicle or 25 μg K1K1. (**A**) Chronic treatment with K1K1 does not change p-AKT/AKT levels while increases the p-ERK/ERK activation in 140 days old SOD1G93A mice with respect to vehicle treated mice. Bars represent mean ± SEM (*n* = 4 − 5). All data were statistically analyzed using Student’s *t*-test *** = *p* < 0.001.

**Figure 4 ijms-21-08542-f004:**
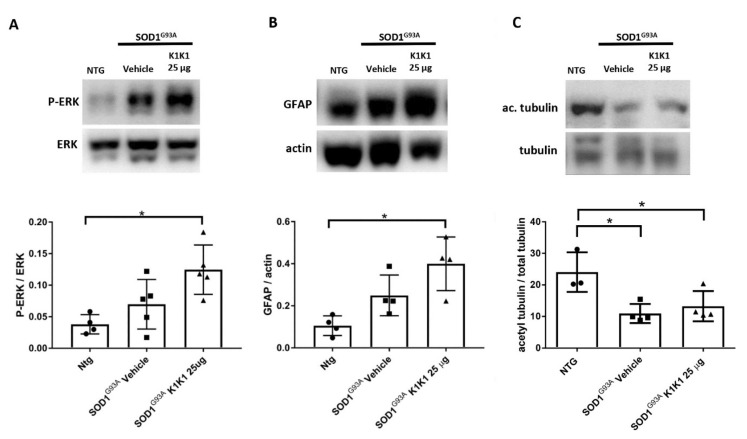
Treatment with 25 μg K1K1 increased ERK phosphorylation and Glia Acid Fibrillary Protein (GFAP) expression in the sciatic nerve of SOD1G93A mice. (**A**) p-ERK levels were mildly higher in SOD1G93A mice treated with vehicle compared to NTG mice and treatment with 25 μg K1K1 significantly increased the phosphorylation of ERK in SOD1G93A mice. A similar effect was observed for the levels of GFAP (**B**). The levels of acetyl-tubulin (**C**) were significantly reduced in SOD1G93A mice, an effect not modified by the K1K1 treatment. Bar graphs represents mean ± SEM (*n* = 4–5). All data were statistically analyzed using One Way ANOVA followed by post hoc Fisher’s least significant difference (LSD). * *p* < 0.05.

**Figure 5 ijms-21-08542-f005:**
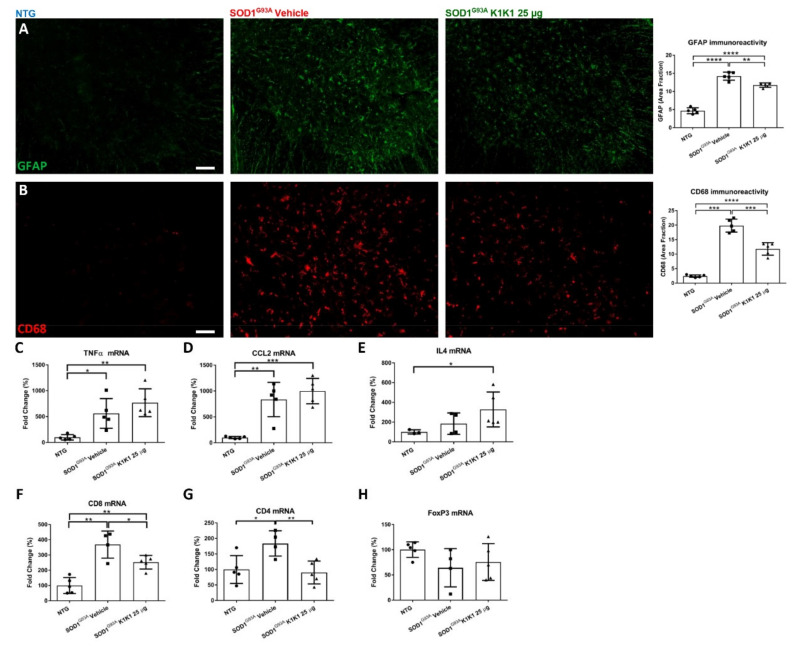
Treatment with 25 μg K1K1 reduces astrocytosis and macrophagic microglia and increases interleukin (IL)-4 in the LSC of SOD1G93A mice. (**A**) Representative images of LSC ventral horn of non-transgenic (NTG) and SOD1G93A mice treated with vehicle or 25 μg K1K1 at the symptomatic stage of the disease (140 days), stained for GFAP (green), scale bar: 50 μm. Quantification of GFAP immunofluorescence showed elevated astrocytosis in the LSC ventral horn of SOD1G93A mice but treatment with K1K1 significantly reduced the reactive astrocytosis. (**B**) Cluster of differentiation (CD)68 positive macrophagic microglia (red) were labelled in ventral horn of LSC, scale bar: 50 μm. The quantification of immunoreactivity showed an increase in CD68^+^ cells in transgenic mice treated with vehicle compared to NTG. This effect was significantly reduced by the treatment with 25 μg K1K1. Data are expressed as mean ± SEM, *n* = 5 animals per group. (**CE**) Real time PCR for pro-inflammatory (Tumor Necrosis Factor (TNF)α and (C-C motif) ligand 2 (CCL2)) and anti-inflammatory (IL-4) marker in the ventral portion of LSC of NTG and transgenic mice (140 days of age), treated with vehicle or 25 μg K1K1. For TNFα and CCL2, the increase observed in SOD1G93A mice treated with vehicle was unchanged by 25 μg K1K1. On the contrary, for IL-4 we observed an increase in 25 μg K1K1 treated mice suggesting an anti-inflammatory response. (**F**–**G**) 25 μg K1K1 reverted the increase in CD8^+^ (**F**) and CD4^+^ T (**G**) cells in lumbar spinal cord of SOD1G93A mice (140 days) while levels of forkhead box P3 (FOXP3) (**H**) that tend to be lower in vehicle SOD1G93A were unmodified by K1K1 treatment. Bars represent the mean ± SEM, *n* = 5 animals for each group. All data were statistically analyzed using One-way ANOVA followed by post hoc Fisher’s LSD. * *p* < 0.05, ** *p* < 0.01, *** *p* < 0.005, **** *p* < 0.001, *n*.s. = not significant.

**Figure 6 ijms-21-08542-f006:**
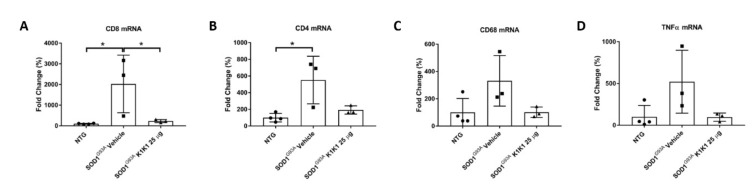
Treatment with 25 μg K1K1 dampens the inflammatory response in Tibialis Anterior muscles of SOD1G93A mice. (**A**–**D**) Real time PCR shows that 25 μg K1K1 decreases CD8^+^ (**A**) and CD4^+^ T (**B**) cells in Tibialis Anterior Muscle. For the mediators of inflammation CD68 (**C**) and TNFa (**D**) we observe a reduction after treatment with 25 μg K1K1. Data are expressed as the mean ± SEM as percentage of NTG (*n* = 3–5 mice per group). All data were statistically analyzed using One-way ANOVA followed by post hoc Fisher’s LSD. * *p* < 0.05.

**Figure 7 ijms-21-08542-f007:**
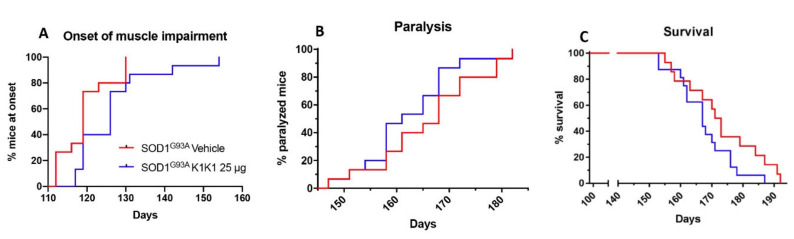
25 μg K1K1 delays neuromuscular impairment but not survival of SOD1G93A mice. (**A**) Kaplan–Meier curve showing, in a second group of mice, that treatment with 25 μg K1K1 significantly delays the onset of motor symptoms (*p* < 0.05, Log-rank test). (**B**,**C**) Kaplan–Meier curve showing that treatment with 25 μg K1K1 did not affect the age of paralysis (*p* = 0.2956), considered as the age at which the mice were no longer able to perform the grip strength test, nor the survival length (*p* = 0.1048) in SOD1G93A mice. The curves were evaluated using the Log-rank test (*n* = 15 animals for each group).

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
