# Peer review of "A Novel HGF/SF Receptor (MET) Agonist Transiently Delays the Disease Progression in an Amyotrophic Lateral Sclerosis Mouse Model by Promoting Neuronal Survival and Dampening the Immune Dysregulation"

_ijms, 2020, doi:10.3390/ijms21228542_

Round 1
Reviewer 1 Report
The article titled “A novel HGF/SF receptor (MET) agonist transiently delays the disease progression in an amyotrophic lateral sclerosis mouse model by promoting neuronal survival and dampening the immune dysregulation” by Vallarola et al. aims to investigate The molecular mechanisms involved in the effects of K1K1 in vivo models of ALS.
Importantly they note the limitations of their findings in the discussion so do not over interpret (which is important in all of these studies). The work is interesting and will be of interest to readers of JMS. I do have a few comments.
Comments to be addressed:
Authors reported in line 139 that “Mice were treated for 6 weeks from the onset of the symptoms until the symptomatic stage” it could be better to indicate the day in which mice were treated IP with PBS as vehicle or mouse K1K1.
In figure 3 are not reported the internal standard for western blotting, as well as in figure 4 (A) and (B), figure A5.
Author Response
Point 1: Authors reported in line 139 that “Mice were treated for 6 weeks from the onset of the symptoms until the symptomatic stage” it could be better to indicate the day in which mice were treated IP with PBS as vehicle or mouse K1K1.
Response 1: Mice have been treated starting from 98 days to 140 days of age. This has been added in methods (parag 2.3 line 150)
Point 2: In figure 3 are not reported the internal standard for western blotting, as well as in figure 4 (A) and (B), figure A5
Response 2: In this case since we were interested in the analysis of the ratio between the modified form of the proteins (phosphorylated for ERK and AKT or acetylated for Tubulin) ) versus total protein, we evaluated that the loading of the total proteins for each sample was comparable using the Ponceau staining. We have not examined an internal standard.
Reviewer 2 Report
All abbreviations including gene names should be introduced in full upon first mention in both the Abstract and main text.
in vitro and in vivo should be in italics
In the caption of Fig 1 "10-10 M" should be "10-10 M"
Minor English corrections are required including:
"This result comply" should read "This result complies"
"at earlier time point" should read "at an earlier time point"
"in presence of attenuated" should read "in the presence of attenuated"
"Ntg" should read "NTG"
"the proportion of Treg" should read "the proportion of Tregs"
"in attempt to preserve" should read "in an attempt to preserve"
"following a damage" should read "following damage"
"Few days after plating, they show" should read "A few days after plating, they showed"
Author Response
Point 1: All abbreviations including gene names should be introduced in full upon first mention in both the Abstract and main text.
- in vitro and in vivo should be in italics
- In the caption of Fig 1 "10-10 M" should be "10-10 M"
- Minor English corrections are required including:
- "This result comply" should read "This result complies"
- "at earlier time point" should read "at an earlier time point"
- "in presence of attenuated" should read "in the presence of attenuated"
- "Ntg" should read "NTG"
- "the proportion of Treg" should read "the proportion of Tregs"
- "in attempt to preserve" should read "in an attempt to preserve"
- "following a damage" should read "following damage"
- "Few days after plating, they show" should read "A few days after plating, they showed"
Response 1: We edited and corrected all the reported errors in the text.
Reviewer 3 Report
In this manuscript from Vallarola et al, the authors describe the therapeutic potential of a novel recombinant HGF/SF receptor agonist, termed K1K1. The authors use a combination of in vivo and in vitro approaches and demonstrate that K1K1 treatment of pre/early symptomatic mice delays disease onset, NMJ degeneration, and reduces neuroinflammation via the activation of the ERK pathway. Despite these initial beneficial effects, long term treatment of SOD1G93A mice fails at increasing survival.
Overall, the study is well designed, and the results are convincing and well presented. One thing that the authors do not address is whether K1K1-dependent activation of the ERK pathway happens broadly in all cell types or whether it is restricted to a subset of cells. While Figure 3 shows the effects of K1K1 treatment of pERK levels, this analysis was performed on whole tissue lysates. However, most of the effects reported in this study are related to the modulation of neuroinflammation, in the spinal cord as well as in the nerve and muscle tissue. Thus, it would be interesting to define whether K1K1 treatment is able to induce ERK activation in microglia as well as in spinal motor neurons. A cell specific effect of K1K1 could at least partially explain why no survival improvement was observed in the SOD1G93A mice. The authors could consider additional immunohistochemistry experiments to more specifically assess K1K1 target engagement in different cell types in the spinal cord.
Author Response
Point 1: In this manuscript from Vallarola et al, the authors describe the therapeutic potential of a novel recombinant HGF/SF receptor agonist, termed K1K1. The authors use a combination of in vivo and in vitro approaches and demonstrate that K1K1 treatment of pre/early symptomatic mice delays disease onset, NMJ degeneration, and reduces neuroinflammation via the activation of the ERK pathway. Despite these initial beneficial effects, long term treatment of SOD1G93A mice fails at increasing survival.
Overall, the study is well designed, and the results are convincing and well presented. One thing that the authors do not address is whether K1K1-dependent activation of the ERK pathway happens broadly in all cell types or whether it is restricted to a subset of cells. While Figure 3 shows the effects of K1K1 treatment of pERK levels, this analysis was performed on whole tissue lysates. However, most of the effects reported in this study are related to the modulation of neuroinflammation, in the spinal cord as well as in the nerve and muscle tissue. Thus, it would be interesting to define whether K1K1 treatment is able to induce ERK activation in microglia as well as in spinal motor neurons. A cell specific effect of K1K1 could at least partially explain why no survival improvement was observed in the SOD1G93A mice. The authors could consider additional immunohistochemistry experiments to more specifically assess K1K1 target engagement in different cell types in the spinal cord.
Response 1: We are in complete agreement with the reviewer and in fact it is our intention to investigate deeper this interesting aspect ... however, unfortunately at this moment we no longer have samples available to perform the analysis.